# Failure Prediction from Few Expert Demonstrations

**Anjali Parashar**,* **Kunal Garg, Joseph Zhang, Chuchu Fan**
Massachusetts Institute of Technology
Email:{anjalip,kgarg,jzha,chuchu}@mit.edu

## Abstract

This extended abstract presents a novel three-step methodology for discovering failures that occur in the true system by using a combination of a minimal number of demonstrations of the true system and the failure information processed through sampling-based testing of a model dynamical system. The proposed methodology comprises a) exhaustive simulations for discovering failures using model dynamics; b) design of initial demonstrations of the true system using Bayesian inference to learn a GPR-based failure predictor; and c) iterative demonstrations of the true system for updating the failure predictor. As a demonstration of the presented methodology, we consider the failure discovery for the task of pushing a T block to a fixed target region with UR3E collaborative robot arm using a diffusion policy and present the preliminary results on failure prediction for the true system.

## 1   Introduction

Testing and validation are essential tools to ensure the safety of autonomous systems prior to deployment [1, 2, 3, 4]. Most of the state-of-the-art tools for model-based validation and falsification of the autonomous system assume access to the true system [5]. These model-based tools mainly use sampling-based methods for failure discovery [6, 7, 8]. While sampling-based techniques allow efficient exploration of the search-space, they require large number of samples to work efficiently, and are therefore well suited for simulation based testing. Most of these approaches assume that the model dynamics and simulation testing environment adequately represent realistic testing conditions. However, this can be misleading, since a sim-to-real gap can lead to unexpected failures that were unobserved in the simulation environment on which the policy was trained [9]. Additionally, uncertainties in state estimation and dynamics can also affect the performance of the autonomous systems. Collectively, these issues lead to failure modes that remain undiscovered, despite exhaustive simulation testing. The resulting sim-to-real gap is especially concerning from the perspective of safety, as the discovered failure modes in simulation may not reflect the true severity of real failures, i.e. a failure not reported as unsafe behavior in simulation may be unsafe and catastrophic for the true system. In this study, we analyze the sim-to-real gap from the perspective of falsification, using a sampling-based testing pipeline for simulation for efficient exploration of failures, while working with limited data from the true system to enable better prediction of failures.

## 2   Problem formulation

Consider a discrete-time closed-loop dynamics:
$$x_{t+1} = f(x_t, \pi(y_t, z)) + \epsilon_1, \qquad y_t = Cx_t + \epsilon_2, \tag{1}$$
where $f : \mathbb{R}^n \times \mathbb{R}^m \to \mathbb{R}^n$ and $C \in \mathbb{R}^{l \times n}$, with state $x_t \in \mathbb{R}^n$ at time $t \in \mathbb{R}$ and policy $\pi : \mathbb{R}^l \times \mathbb{R}^d \to \mathcal{U} \subseteq \mathbb{R}^m$ which outputs actions based on environmental variables $z \in \mathcal{Z} \subset \mathbb{R}^d$ and system

---

*Anjali Parashar is the corresponding author. Project website: https://mit-realm.github.io/few-demo/

Workshop on Bayesian Decision-making and Uncertainty, 38th Conference on Neural Information Processing Systems (NeurIPS 2024).

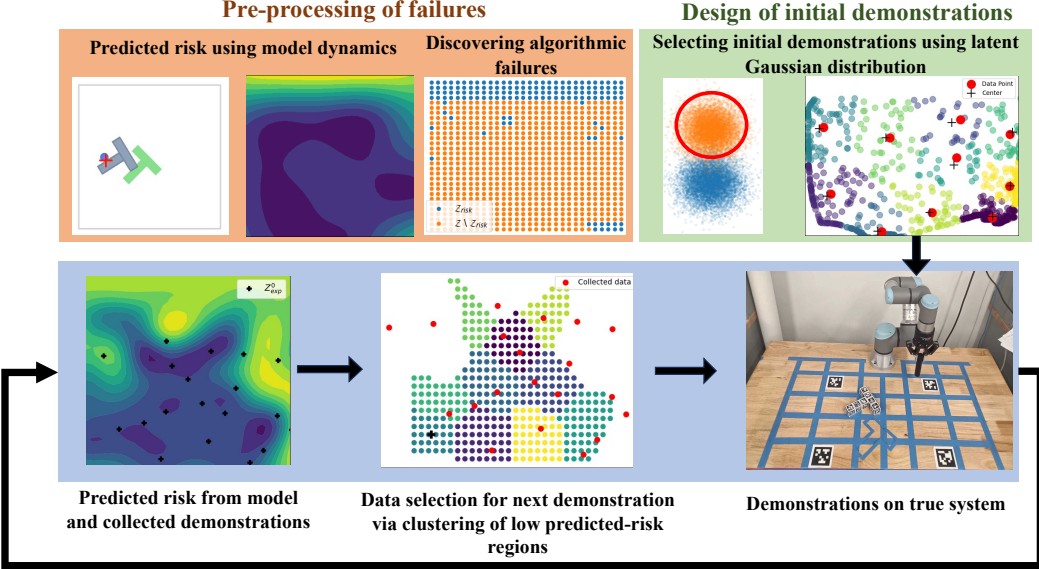

**Figure 1:** The proposed methodology constitutes a) discovering failures using model information; b) design of initial demonstrations to learn true system failures using Bayesian inference; and c) sequential demonstration from low predicted-risk regions for GPR-based risk prediction update.

output $y_t \in \mathbb{R}^l$, where $\epsilon_1$ and $\epsilon_2$ are disturbances in dynamics and state estimation, respectively. The environment variable $z$ represents variables that can be independently controlled by the user, such as initial conditions of the system $x_0$, and environmental information exogenous to the agent. In this work, we assume that the disturbances come from zero-mean Gaussian distributions given by $\epsilon_1 \sim \mathcal{N}(\mathbf{0}, \Sigma_1)$ and $\epsilon_2 \sim \mathcal{N}(\mathbf{0}, \Sigma_2)$, where the covariance matrices $\Sigma_1 \in \mathbb{R}^{n \times n}, \Sigma_2 \in \mathbb{R}^{l \times l}$ of the distributions are defined using scalars $\sigma_1, \sigma_2 > 0$ as $\Sigma_i = \sigma_i \mathbf{I}$, for $i = 1, 2$, where $\mathbf{I}$ is an identity matrix of the appropriate size. We consider two set of dynamics in this paper: model (known to the user) and true (unknown to the user). *True dynamics* corresponds to the actual dynamics of the agent, which is unknown whereas *Model dynamics* corresponds to the estimate of the true dynamics and is described as in (1). We denote a trajectory rollout of (1) for a given environment variable $z$ under a given $(\sigma_1, \sigma_2)$ as $X_{(z|\sigma_1,\sigma_2)} = (x_i)_{i=0}^T$.[2] The trajectory rollout of the true dynamics for a given environment variable $z$ is denoted as $X_z^*$.

In this work, we address the problem of discovering failures of the true dynamical system with limited demonstrations. For this purpose, we consider a user-defined risk function $R : \mathbb{R}^d \to \mathbb{R}$ where $R(z) = R(z, X_z)$ denotes the risk corresponding to the trajectory rollout $X_z$ for a given environment variable $z$. Based on this risk function, we define failure of the system when the risk $R(z)$ for a corresponding $z$ exceeds a user-defined threshold $R_{\mathrm{th}} \in \mathbb{R}$. As thus, the falsification problem can be mathematically formulated as discovering the set $\mathcal{Z}_{\mathrm{fail}}^* := \{z \mid R(z, X_z^*) > R_{\mathrm{th}}\}$. We aim to solve this under the constraint that we can query the true system only a few times $N > 0$ to obtain $N$ trajectory rollouts $\{X_{z_i}^*\}_{i=1}^N$ for $z_i \in \mathcal{Z}$. We present a three-step methodology to discover failures occurring in the true system by using a combination of a minimal number of demonstrations $\{X_z^*\}$ and the failure information from the model dynamics obtained through sampling-based falsification.

## 3 Methodology

We assume that the model can capture a subset of the failures that could occur with the true system, i.e., $\mathcal{Z}_{\mathrm{fail}}(f) \cap \mathcal{Z}_{\mathrm{fail}}^* \neq \emptyset$. Based on this assumption, we obtain that the $\mathcal{Z}_{\mathrm{fail}}^* \subseteq \mathcal{Z}_{\mathrm{fail}}(f) \cup \mathcal{Z}_{\mathrm{real}}$, i.e, failures of the true system are a combination of algorithmic failures on the model system $\mathcal{Z}_{\mathrm{fail}}(f)$ and failures due to the mismatch between model dynamics and actual dynamics, disturbances and other unknown reasons $\mathcal{Z}_{\mathrm{real}}$. We say that the set $\mathcal{Z}_{\mathrm{fail}}(f)$ captures algorithmic failures as we assume that

---

[2]In what follows, we suppress the explicit dependence on $\sigma_1, \sigma_2$ for the sake of brevity.

the policy $\pi$ is trained for the model $f$ but still leads to failures. The first step of our methodology focuses on identifying failures that can be obtained using the model information.

### 3.1 Pre-processing of failures: utilizing model information

We define the set of the environment variables for the algorithmic failures as:

$$\mathcal{Z}_{\text{fail}}(f) := \{z \mid R(z, X_{(z|\sigma_1=0,\sigma_2=0)}) > R_{\text{th}}\}, \tag{2}$$

which captures the algorithmic failure of the model dynamics. This set can be readily obtained through extensive simulations using the model information. Next, we aim to capture the failures due to the mismatch between model dynamics and true dynamics, disturbances, and potentially other unknown reasons. For this, we sample $\sigma_1, \sigma_2$ from a bounded region given by $\mathcal{B} \in [\sigma_1^{\min}, \sigma_1^{\max}] \times [\sigma_2^{\min}, \sigma_2^{\max}]$, and observe the risk $R(z, X_z)$ corresponding to each disturbance, and collect values of $z$ for which a failure is observed across all disturbances:

$$\mathcal{Z}_{\text{noise}} := \{z \mid R(z, X_{(z|\sigma_1,\sigma_2)}) > R_{\text{th}} \ \forall \ \sigma_1, \sigma_2 \in \mathcal{B} \setminus \{(0,0)\}\}, \tag{3}$$

so that $\mathcal{Z}_{\text{noise}} \subset \mathcal{Z}_{\text{real}}$. The region $\mathcal{Z}_{\text{risk}} := \mathcal{Z}_{\text{noise}} \cup \mathcal{Z}_{\text{fail}}(f)$ captures failures that can be discovered using model dynamics. Next, we aim to discover failures that the model system cannot capture through sampling $z$ from the region $\mathcal{Z}_{\text{real}}$ and obtaining demonstrations from true dynamics. Since $\mathcal{Z}_{\text{real}}$ is not known, we obtain these samples from the region $\mathcal{Z} \setminus \mathcal{Z}_{\text{risk}}$ as discussed in the next section.

### 3.2 Sampling from sensitive regions: Design of experiments

Since we have a limited budget on the number of demonstrations we can obtain from the true dynamics, we aim to maximize the state-space covered in each of these demonstrations. For a given $z$, we define a coverage function, given by $C : \mathcal{Z} \to \mathbb{R}$ given as $C(z) = C(z, X_z)$, which is a monotonic function of the state-space explored along the trajectory $X_z$, and aim to sample $z \in \mathcal{Z} \setminus \mathcal{Z}_{\text{risk}}$ from a distribution $\mathbb{P}$ corresponding to high coverage:

$$z \sim \mathbb{P}(z \mid C(z, X_z) > C_{\text{th}}, z \in \mathcal{Z} \setminus \mathcal{Z}_{\text{risk}}), \tag{4}$$

where $C_{\text{th}} > 0$ is a user-defined coverage threshold. Directly sampling from this distribution is intractable, so we use a Bayesian inference framework [7] with the posterior distribution as defined in (4). To ensure that the generated samples lie in the region $\mathcal{Z} \setminus \mathcal{Z}_{\text{risk}}$, we use a Normalizing Flows based framework for classification, called Flow-GMM [10] to learn Gaussian distributions in latent space $\mathcal{W} \subseteq \mathbb{R}^d$ corresponding to the sets $\mathcal{Z}_{\text{risk}}$ and $\mathcal{Z} \setminus \mathcal{Z}_{\text{risk}}$, given by $\tilde{p}_1$ and $\tilde{p}_2$ respectively, and reconstruct the Bayesian inference in $\mathcal{W}$. We then use Metropolis-Hastings algorithm [11] to sample from the defined posterior distribution and apply a projection operator on the generated samples to ensure that we sample exclusively from $\tilde{p}_2$. The details of posterior construction and projection can be found in Appendix B. The pipeline discussed so far generates a collection of samples $Z_{\text{cov}} = \{z \in \mathcal{Z} \setminus \mathcal{Z}_{\text{risk}} \mid C(z, X_z) > C_{\text{th}}\}$. Once we have generated the samples, we choose $N/2$ candidate values of $z$ distributed uniformly across the search-space. This is achieved by dividing $Z_{\text{cov}}$ into $N/2$ clusters using K-means clustering, and choosing the geometric centers of the generated clusters for demonstrations. This allow us to collect $N/2$ data points $Z_1 = \{z_j\}$ with corresponding risk values given by $R_1 = \{R(z_j, X_{z_j}^*)\}$. We also obtain $M$ data points $Z_2 = \{z_i\}$ from the region $\mathcal{Z}_{\text{risk}}$, with the corresponding risk values given by $R_2 = \{R(z_i, X_{z_i})\}$ using model dynamics $f$. We define $\mathcal{D}_1 = [Z_1, R_1]$ and $\mathcal{D}_2 = [Z_2, R_2]$ as the dataset of demonstrations obtained from true and model system, respectively, and use them to train a model $\phi_\theta : \mathcal{Z} \to \mathbb{R}$ to predict risk $\hat{R} = \phi_\theta(z)$ for a given $z$, as illustrated in the next section where $\theta$ denotes the model parameters.

### 3.3 Sequential failure prediction and training using Gaussian Processes

Motivated by the success of Gaussian Process Regression (GPR) in learning from limited demonstrations [12], we use GPR as the backbone of the risk prediction pipeline in this section. Using the dataset $\mathcal{D}_f = \mathcal{D}_1 \cup \mathcal{D}_2$, we construct the marginal log likelihood $\log p_{\phi_\theta}(R_f | Z_{\text{f}})$ for learning the model $\phi_\theta$ using a sum of the marginal log likelihoods from both sources of data as:

$$\log p_{\phi_\theta}(R_f | Z_{\text{f}}) = \log p_{\phi_\theta}(R_1 | Z_1) + \log p_{\phi_\theta}(R_2 | Z_2), \tag{5}$$

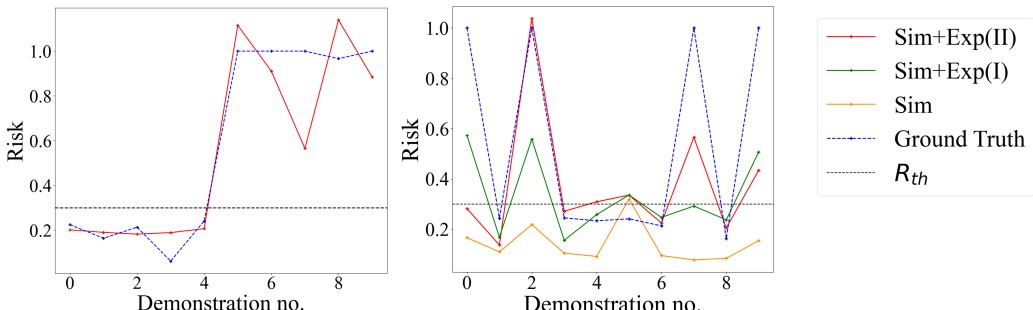

**Figure 2:** The left plot demonstrates model prediction on 10 data points where the predicted risk by the learned model $\phi_\theta$ is either very low ($R < 0.3$) or very high ($R > 0.3$). The demonstration on true system marked as 'Ground Truth' illustrates the prediction to be accurate. The right plot shows the prediction of models on 10 random data points with data collected using different methods considered in Table. 1, compared against the risk observed from demonstrations.

where $Z_f = [Z_1, Z_2]$ and $R_f = [R_1, R_2]$. The training objective can be formulated as the maximization of the marginal log likelihood in (5) with $\phi_\theta$ as the decision variable:

$$\phi_{\theta^*} = \arg\max_\theta \log p_{\phi_\theta}(R_f | Z_f). \tag{6}$$

We first learn a model $\phi_{\theta_0}$ for risk prediction using $\mathcal{D}_1$ and $\mathcal{D}_2$ and subsequently refine the model by a sequence of $N/2$ demonstrations on the true system with sequential optimization of $\phi_\theta$ solving (6) with the updated dataset and generation of data-point for the next demonstration. The details of sequential demonstration and risk prediction update can be found in Appendix C.

## 4    Falsification of diffusion policy on Push-T

As a demonstration of the our methodology, we consider the task of pushing a T-block to a fixed target region with a circular end-effector using a diffusion policy from [13] which predicts actions conditioned on observations. Fig. 4 in Appendix D shows the setup corresponding to the model and true system respectively. Appendix D has details of the model dynamics and experimental setup.

We restrict the number of hardware demonstrations to $N = 20$. Fig. 1 (see the plot under 'Discovering algorithmic failures') shows the region $\mathcal{Z}_{\text{risk}}$ discovered in simulation using the model dynamics. For validating the learned failure prediction using the learned model $\phi_\theta$, we record the risk for 10 randomly sampled test demonstrations on the the true system, and compare against the predictions from our method and two other baselines. Table. 1 shows the risk prediction error with GPR using three methods, namely, data collected only using model dynamics (reported as Simulation), data corresponding to $\mathcal{Z}_{\text{risk}} \cup \mathcal{Z}_{\text{fail}}$ from model dynamics and uniformly chosen $N$ data points on the true system (reported as Simulation+Exp (I)), and data collected using our approach (reported as Simulation+Exp (II)). The individual predictions from all three approaches and their comparison against the ground truth (risk from true system) is shown in the right plot in Fig. 2. We observe that the mean prediction error decreases with the chosen data collection scheme, however, the maximum error is higher for our approach, when compared to Simulation + Exp (I). This is due to the fact that the uniform sampling was able to discover failures that were unobserved with our method due to lack of sufficient exploration. To address this issue, we aim to examine a combination of exploration and exploitation in the data collection schemes. Appendix D provides a detailed analysis of the results summarized in this section.

**Table 1:** Failure prediction baseline comparison

|                        | Mean Error | Max Error | Std. Deviation |
| ---------------------- | ---------- | --------- | -------------- |
| Simulation             | 0.38       | 0.85      | 0.33           |
| Simulation + Exp (I)   | 0.23       | 0.54      | 0.2            |
| Simulation + Exp (II)  | 0.21       | 0.72      | 0.25           |

We also validate the learned risk by conducting demonstrations on 10 data points sampled from predicted high-risk and predicted low-risk regions. The predicted and ground truth values of risk corresponding to these points is shown in the left plot in Fig. 2. As we can see, the GPR model accurately predict 'fail' and 'not fail' across all 10 data points, where failure for a chosen value of $z$ corresponds to the predicted risk being higher than the threshold $R_{\text{th}} = 0.3$. The complete set of results and hardware demonstrations for the Push-T task and additional tasks considered in this work can be found at the project website [3]

## 5    Conclusion

In this paper, we present a novel scheme for discovering failures that occur due to sim-to-real gap using Bayesian inference principles. The pipeline presented for initial estimation of failures from model dynamics in Section 3.1 is built on Bayesian inference principles and leverages the expressivity of sampling techniques, which is followed by a sequential failure prediction pipeline. The approach for sequential failure prediction presented in Section 3.3 can be expressed more formally through Bayesian Experimental Design (BED) [14], which comprises future scope of work. The usage of Gaussian Processes for failure learning as a model choice works well with limited demonstration setting, however, poses scalability challenges, since the number of demonstrations required increases with the dimension of search space. We aim to address this in the future work, potentially by leveraging data-efficient and scalable models such as Variational Gaussian Processes (VGP) [15].

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

# A  Related work

Testing and model validation are are essential tools for ensuring safety of autonomous systems before deployment. There are several model-based and model-agnostic tools for testing and falsification that have been proposed in the literature [1, 2, 3, 4, 5]. The model-based tools have the advantage of being faster [6, 8], and with differentiable system models, gradient-based optimization strategies can be used for falsification, which have shown to be more efficient than black-box methods [8] and adversarial optimization [16, 17]. The model-agnostic tools on the other hand, do not rely on model information to discover failure modes [18, 19].

Recently, some works have considered simulation-based falsification and testing [8, 20]. These methods minimize the dependency on hardware-based for testing, which can be extremely time-consuming and require extensive utilization of resources [21]. Most of these approaches assume that the simulated dynamic system and testing environment represent the realistic testing conditions adequately. However, this can be misleading, especially true for the case of learning-based policies, where a sim-to-real gap can lead to unexpected failures that were unobserved in simulation environment, on which the policy was trained [9, 22]. Additionally, there are uncertainties in state estimation and dynamics, that affect the performance of cyber-physical systems. Collectively, these issues leads to failure modes that remain undiscovered, despite exhaustive simulation testing, due to the inadequacy of the simulation models to capture wide-ranging practical phenomenon that affect the real-word system.

In this study, we analyze the sim-to-real gap from the perspective of falsification, using existing testing pipelines for simulation, while working with limited real-time data to enable better prediction of failures.

# B  Sampling from latent distribution with constraints

In (4), we use population variance of the trajectory rollout $X_z$ as a coverage metric $C$. To sample exclusively from $\mathcal{Z}\setminus\mathcal{Z}_{\text{risk}}$, we learn the decision boundary that distinguishes $\mathcal{Z}_{\text{risk}}$ from the remaining search space by performing supervised binary classification using samples collected in the previous step. For this, we use a specific technique known as Flow-GMM [10], which allows us to learn an invertible mapping from search-space $\mathcal{Z}$ to a latent-space $\mathcal{W} \in \mathbb{R}^d$, given by $g_\theta^{-1} : \mathcal{Z} \to \mathcal{W}$ using the Normalizing Flows framework [23]. Using Flow-GMM, we learn isotropic Gaussian latent distributions corresponding to the base distributions $p_1 = p_{\mathcal{Z}_{\text{risk}}}$ and $p_2 = p_{\mathcal{Z}\setminus\mathcal{Z}_{\text{risk}}}$ which can be mathematically expressed using mean $\mu_i^g$ and covariance $\Sigma_i^g$ as $\tilde{p}_i = \mathcal{N}(\mu_i^g, \Sigma_i^g)$ for $i = 1, 2$.

Sampling directly from the distribution generated by (4) is intractable, hence, we utilize a Bayesian inference framework here. Additionally, we make use of the learnt mapping $g_\theta^{-1} : \mathcal{Z} \to \mathcal{W}$ to simplify the posterior for sampling in the latent space $\mathcal{W}$:

$$w \sim p(w|C(g_\theta(w), X_{g_\theta(w)}) > C_{\text{th}}, \text{label} = 2) \tag{7}$$

The conversion of distribution from (4) to (7) is done because sampling $z$ from the base distribution $p_2$ can be challenging as $p_2$ might be multi-modal. However, the corresponding latent distribution $\tilde{p}_2$ is unimodal, thereby allowing us to apply projection using a projection operator $\mathbf{P}[w]$ to ensure that a generated sample $w$ remains within a prescribed convex set centered at the mean $\mu_2^g$ of $\tilde{p}_2$. This ensures that samples are only drawn from the set $\mathcal{Z}\setminus\mathcal{Z}_{\text{risk}}$. The construction of convex boundary and the corresponding projection operator has been discussed in the next section. We utilize exponential modelling for expressing the likelihood $p(C > C_{\text{th}}|g_\theta(w))$, adopted from [7] and $\tilde{p}_2$ as the prior to construct the posterior for sampling. The sampled latent environment variable $\tilde{w}$ is obtained as $\tilde{w} = \mathbf{P}[w]$ where:

$$w \sim p(w|C(g_\theta(w), X_{g_\theta(w)}) > C_{\text{th}}) \propto \exp\left(C(g_\theta(w), X_{g_\theta(w)})\right)\tilde{p}_2(w). \tag{8}$$

We use Metropolis Hashtings to sample $w$ from the constructed posterior distribution, which is a gradient-free sampling method, as application considered in this paper corresponds to a non-differentiable dynamic system (Section-4). The pipeline discussed so far generates a collection of samples $Z_{\text{cov}} = \{z \in \mathcal{Z}\setminus\mathcal{Z}_{\text{risk}} \mid C(z, X_z) > C_{\text{th}}\}$.

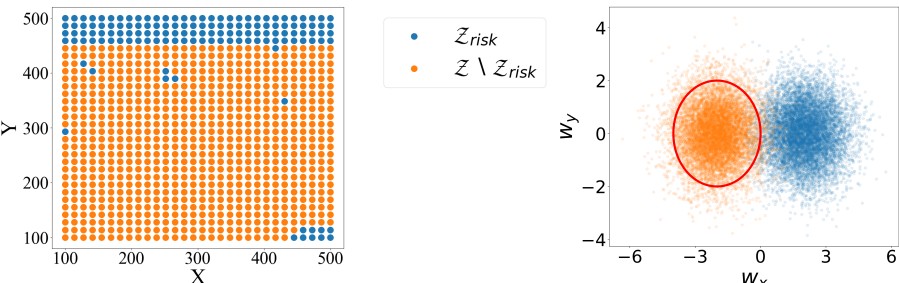

**Figure 3:** The left plot shows region $\mathcal{Z}\backslash\mathcal{Z}_{\text{risk}}$ in orange, discovered using the methodology described in Section 3.1. The right plot shows the latent distributions $\tilde{p}_1$, $\tilde{p}_2$ in blue and orange respectively. To sample exclusively from $\tilde{p}_2$, we project the samples within the region $\mathcal{P}$ with the boundary shown in red

### B.1 Sampling with constraints: construction of projection operator

While we are sampling from the latent distribution $w \sim \tilde{p}_2$, the goal is to sample strictly from the region $\mathcal{Z}\backslash\mathcal{Z}_{\text{risk}}$, which is given by $g_\theta(w)$. For the learnt latent distributions, there exists a set $\mathcal{P}$, such that:

$$\tilde{p}_2(w) > \tilde{p}_1(w) \ \forall \ w \in \mathcal{P}. \tag{9}$$

For a point lying outside this region, we may have $\tilde{p}_1(w) > \tilde{p}_2(w)$, which means that the generated sample corresponds to the region $\mathcal{Z}_{\text{risk}}$. Hence, we use the boundary $\mathcal{P}$ to generate samples such that the corresponding $z$ lies in desired region. We first construct an explicit boundary $\mathcal{P}$ and subsequently utilize it for projection. For an isotropic Gaussian distribution $\mathcal{N}(\mu, \sigma)$, where the covariance is given by $\Sigma = \sigma\boldsymbol{I}$, there exists $r \in \mathbb{R}, c \in \mathbb{R}^d$, such that $\mathcal{P} = \{w|\ \|w - c\|_2^2 \leq r^2\}$ meets the requirement specified in (9). Note that the value of $r$ that satisfies (6) for a given $c$ is not unique, and can be decreased to make the sampling more conservative or vice versa. In our simulations, we chose $c = \mu_2^g$ which is the mean of the Gaussian distribution $\tilde{p}_2$ and a user-defined hyperparameter in the training of Flow-GMM. The chosen $\mathcal{P}$ is convex, for which a projection operator can be constructed as:

$$\boldsymbol{P}[w] = c + r\frac{w - c}{\|w - c\|_2}. \tag{10}$$

It can be easily verified that $\boldsymbol{P}[w]$ lies on the boundary of $\mathcal{P}$, i.e, $\|\boldsymbol{P}[w] - c\|_2^2 = r$, and hence projects any point $w \notin \mathcal{P}$ onto the boundary. Fig. 3 (right) shows the latent distribution learnt using Flow-GMM corresponding to $\mathcal{Z}_{\text{risk}}$ and $\mathcal{Z}\backslash\mathcal{Z}_{\text{risk}}$, and the boundary of the constructed set $\mathcal{P}$ in red. Here, $d = 2$, and $r = 2$.

## C Sequential demonstration and risk prediction update

We first learn a model $\phi_{\theta_0}$ for risk prediction using initially chosen set of $N/2$ demonstrations from the true system and data corresponding to algorithmic failures and noise ($\mathcal{Z}_{\text{risk}}$) from the model. We then refine the model by a sequence of $N/2$ demonstrations on the true system with sequential optimization of $\phi_\theta$ solving (6) and generation of data-point for the next demonstration. At each step $k$, we first divide the region which are predicted 'not fail' by the learned model, i.e., $\phi_\theta(z) < R_{\text{th}}$ into $N/2$ clusters using K-Means clustering. We then choose $z_k$ from the set of geometric means of the clusters given by $\mathcal{C}_k = [c_1, \ldots, c_{N/2}]$ as the point which maximizes the distance from the previously chosen points. This can be expressed mathematically as:

$$z_k = \arg\max_{z \in \mathcal{C}_k} \min_{c \in Z_{k-1}} \|z - c\|_2^2 \quad k = 1, \ldots, N/2 \tag{11}$$

Here $Z_{k-1} = [z_1, \ldots, z_{k-1}]$. Each step of demonstration is followed by retraining of the GPR model $\phi_\theta$ with the dataset updated with $z_k$:

$$\phi_{\theta_k} = \arg\max_\theta \log p(\hat{R}|\phi_\theta, [Z_{\text{f}}, (z)_{i=1}^k]) \tag{12}$$

Since the number of data points is limited, re-training the GPR model is not challenging. For a larger dataset, updating the pre-trained model instead of re-training would be a more computationally efficient.

# D  Experimental setup

We use a UR3E Collaborative Robot Arm equipped with a Robotiq gripper to hold a 3D printed cylinder and T-shaped block for the circular end-effector and the T-block respectively to construct the true system for the Push-T example. The policy was trained in simulation environment using pre-available dataset, where the workspace of the actual robot is not taken into consideration, and the end-effector is assumed to have only 2D motion in the XY plane without any constraints within a designed box. The policy is known to be robust to visual perturbations, and the implemented policy is trained in simulation using a `PyMunk` and `Gym` environment [24]. The model dynamics here represents the interactions of the end-effector and the T-block, and that of the T-block and the table, is non-differentiable and does not take into account the kinematics and dynamics of the manipulator. We implement the learned policy on a hardware setup using the UR3E manipulator.

The action generated by the policy $\pi$ consists of $(x_e, y_e)$, which corresponds to the XY coordinates of the goal position of the circular end-effector. Fig. 4 shows the model dynamics setup used for training and true system for demonstrations for the Push-T task. We used a Move-It based controller for the robot [25], to move the end-effector to the desired goal location generated by the policy. The risk is calculated as the maximum percentage area of the T-block that overlaps with the target region which is fixed across all experiments, and is normalized to remain within $[0, 1]$. For training the Gaussian Process, we utilize the full trajectory data and assume that for a trajectory that leads to failure, every point on the trajectory corresponds to an initial state which will lead to failure. This is done to maximize the amount of information we can obtain from limited demonstrations.

## D.1  Post analysis of results

There are primarily two sources of sim-to-real gap in the considered Push-T example, which give rise to additional failures, namely, the workspace constraints of the UR3E robot, and the self-collisions of the manipulator. Neither of these conditions are accounted for in the simulation environment, and we discover these causes through the sequential demonstrations conducted using our approach in the paper. Fig. 5 shows the risk prediction contour using GPR with our method our method (Sim+Exp(II)) and Sim+Exp(I). Fig. 6 shows the risk prediction contour using data collected on model system only. We discuss two key differences in the predicted risk using our method (Sim+Exp(II)) and Sim+Exp(I) below.

### D.1.1  Randomness in prediction due to workspace limitations

The regions marked in orange in Fig. 5 represent failure regions that occur due to workspace limitations of the robot. Specifically, the robot is unable to go beyond the physical limit of $y = 400$, whereas the end-effector is assumed to move freely till $y = 500$ in the model. Hence, if the policy generates the goal position to be $y_e \geq 400$, the robot cannot move. In such a scenario, we run repeated experiments, and due to the stochasticity of the true system (caused by non-uniform interaction of the T-block and table surface), there is randomness in failure observed for values of $z$ with $z_y \geq 350$. This leads to the datasets Sim+Exp(I) and Sim+Exp(II) having different values of risk corresponding to these values, consequently reflected in the contours, where our method predicts parts of the region for $y \geq 350$ as low risk, whereas Sim+Exp(I) predicts failure for all $y \geq 350$. Our method cannot capture this randomness effectively, and is therefore a limitation of our approach.

### D.1.2  Self-collisions of the robot

Our method always selects the geometric mean of the generated clusters for demonstration, both in the initial demonstrations (Section-3.2) and sequential demonstrations (Section-3.3). While this leads to high coverage and validation of low risk regions, we are unable to sample from the extremities of the workspace. For values of $y \leq 150$, self-collisions are very likely, and are captured by Sim+Exp(I), whereas our method does not have a data point in that region, leading to the low risk prediction in the region marked in blue in Fig. 5. This leads to the high max error observed in Table. 1 using our method. We observe this as another limitation of our presented pipeline, and the data selection for sequential demonstrations can be improved further to incorporate the discussed issues, particularly by considering a formal sensitivity analysis of the data on the model prediction accuracy, which has been explored in [26, 27] and shown to be effective in predicting generalization of data-driven models [28, 29].

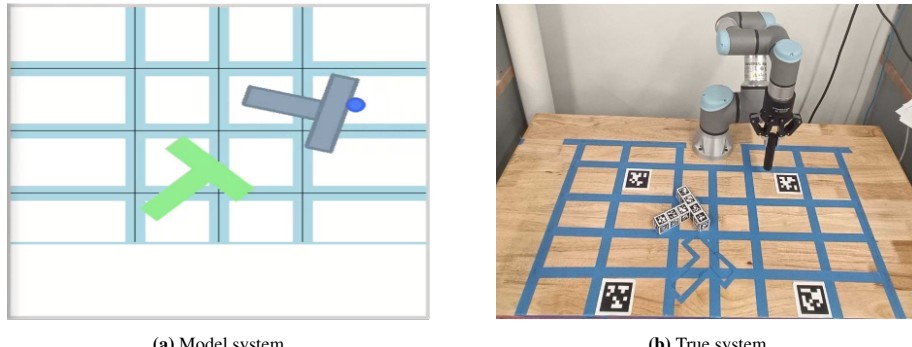

(a) Model system

(b) True system

**Figure 4:** Fig (a) and (b) show the simulation environment and the hardware setup corresponding to the Push-T task considered in Section-4 respectively. The region for demonstrations on true system is shown by the blue boundaries in both figures. In addition to disturbances in state estimation, robot kinematics affects the workspace of the end-effector, leading to failures which are not found in model dynamics.

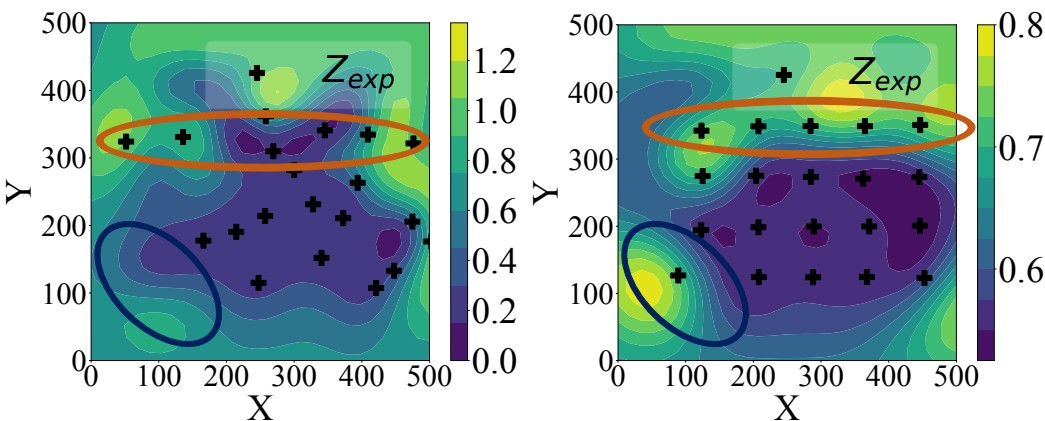

**Figure 5:** Risk Predictions using Sim+Exp (II) (left) and Sim+Exp (I) (right). Areas corresponding to key differences in prediction are highlighted by blue and orange ellipses.

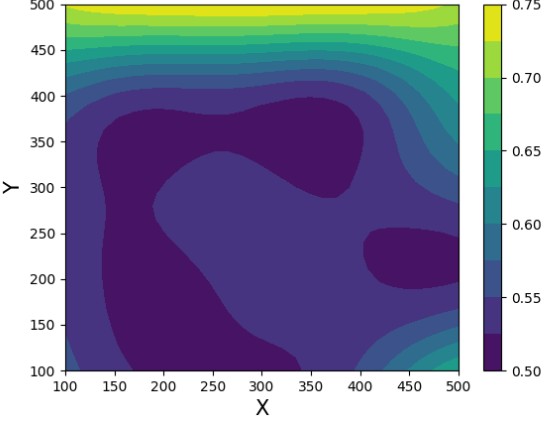

**Figure 6:** Risk Predictions using data from model system only