# OpenReview forum: "Failure Prediction from Few Expert Demonstrations"
_NeurIPS.cc/2024/Workshop/BDU — NeurIPS BDU Workshop 2024 Poster_

### Official Review · Reviewer_kNxh · 2024-09-27
**Very interesting; however, the notation and thinking of edge cases leave something to be desired**

**Rating:** 5
**Confidence:** 2

**Review:**

## Review

- Effectiveness of the coverage function $C(z)$ relies on its ability to capture meaningful exploration of the state space. An improperly defined coverage metric might lead to suboptimal sampling.
- Very interesting application of Bayesian inference using Normalizing Flows (Flow-GMM) to sample $z$ values that are likely to provide high coverage and are outside the known risk regions ($\mathcal{Z}_{\text{risk}}$).
- Scalability questionable: Gaussian Processes work well with small-ish datasets, the computational complexity ($\mathcal{O}(n^3)$) scales poorly with the number of data points. Might be a bottleneck for larger systems requiring more demos.
- Regions where the system's behavior is highly stochastic may be difficult to capture / maximizing coverage may overlook rare but critical failure modes that do not significantly contribute to the coverage metric.

<br>

- I wonder whether using zero-mean Gaussian distributions for $\epsilon_1$ and $\epsilon_2$ is entirely optimal with respect to modeling some approximation of "real-world" disturbances. Also in highly nonlinear systems, even Gaussian input disturbances can lead to non-Gaussian behavior in the state variables.
- Addressing potential sim-to-real gaps (possible lower-fidelity simulations due to assumptions / software used [e.g. PyMunk]) would be beef up the central argument: "Does the software work as intended" risk.
- A very interesting paper overall, a good pitch in my estimation.
- I may lack some mechanical engineering/physics training to fully appreciate the mechanics aspects of the robot arm/robotics.

<br>

- In Table 1, providing uncertainty intervals or detailed error analysis could help clarify why the Max Error increased from "Simulation + Exp (I)" to "Simulation + Exp (II)".
- The paper should provide a clear definition or example of how the coverage function $C(z)$ is computed from a trajectory $X_z$ - important for understanding the sampling strategy
- Also, how is the risk function $R(z)$ calculated? Be nice to have in the appendix at least.
- Should provide additional explanation/examples of how the sampling and projection in the latent space are performed / the transformation to the latent space $W$ using $g_{\theta}^{-1}$ and the use of Normalizing Flows could use a little elaboration
- The projected point $\mathcal{P}[w]$ should satisfy: $\|\mathcal{P}[w] - c\| = r$. Squaring the denominator like in the paper does not result in a unit vector. Instead, seems to me the correct projection operator is: $\mathcal{P}[w] = c + r \frac{w - c}{\| w - c \|}$.

---

### Official Review · Reviewer_DdjB · 2024-10-09
**The paper seems to be somewhat unclear and incomplete.**

**Rating:** 4
**Confidence:** 2

**Review:**

Thanks for submitting this interesting paper.

The overall idea of the paper is good, but it seems to miss some important background information or references for the readers without specific knowledge of the problem setup in the paper to understand.

The abstract for this paper is a bit unclear, and the paper seems incomplete since the last section (section 4) does not have a conclusion or an open discussion.

---

### Official Review · Reviewer_pg4h · 2024-10-09
**Interesting method for efficient failure discovery in true systems**

**Rating:** 6
**Confidence:** 3

**Review:**

This paper paper presents a very interesting and useful methodology for efficiently discovering failures in the true system.

The motivation of the paper is clear as it is obvious to me why practitioners would need to estimate failures with a minimal number of demonstrations of the true system.

The problem formulation clearly defined and easy to understand.

The author's proposed methodology is intuitive and easy to follow. Figure 1 nicely illustrates the proposed three-step pipeline proposed by the authors. The proposed method presents a novel way of utilizing Bayesian inference that I think will be of interest to people at the BDU workshop.

The chosen example of pushing a T block to a fixed target region is interesting and nicely shows the effectiveness of the author's method. However, I note that additional examples would be essential to show the robustness of this method across tasks, and I suggest the authors add additional examples and results before preparing this work for a main conference submission in the future. Additionally, results showing the limitations of this method would be a very useful addition. For example, how does the method perform when 1) the sim-to-real gap becomes larger, 2) the number of of demonstrations of the true system becomes more limited.

Overall, the paper presents an interesting approach applying Bayesian machine learning to solve a relevant problem, and I think the audience at the BDU workshop will find this interesting. I therefore vote to accept the paper. However, in the future significant work remains to be done to strengthen the results and show both the effectiveness and limitations of the proposed method.

---

### Decision · Program_Chairs · 2024-10-09

**Decision:**

Accept (Poster)

**Comment:**

Reviews are mixed. One is very short and has not much content, the other two have relatively low scores but the actual text is more positive. Therefore the scores are likely down to personal reviewing style and not the merit of the work. On this basis, I recommend accepting the work.